# From *cis*-Lobeline to *trans*-Lobeline: Study on the Pharmacodynamics and Isomerization Factors

**DOI:** 10.3390/molecules27196253

**Published:** 2022-09-22

**Authors:** Huan-Hua Xu, Liang Yang, Ming-Xia Tang, An-Ping Ye, Bo-Dan Tu, Zhen-Hong Jiang, Jian-Feng Yi

**Affiliations:** 1Key Laboratory of Modern Preparation of TCM, Ministry of Education, Jiangxi University of Chinese Medicine, Nanchang 330004, China; 2School of Chemistry and Chemical Engineering, Hefei Normal University, Hefei 230601, China; 3Jiangxi Province Key Laboratory of Molecular Medicine, Nanchang 330006, China; 4Research Center for Differentiation and Development of Traditional Chinese Medicine Basic Theory, Jiangxi University of Chinese Medicine, Nanchang 330004, China

**Keywords:** lobeline, isomer, pharmacodynamic effect, isomerization factor

## Abstract

Lobeline is an alkaloid derived from the leaves of an Indian tobacco plant (*Lobelia inflata*), which has been prepared by chemical synthesis. It is classified as a partial nicotinic agonist and has a long history of therapeutic usage ranging from emetic and respiratory stimulant to tobacco smoking cessation agent. The presence of both *cis* and *trans* isomers in lobeline is well known, and many studies on the relationship between the structure and pharmacological activity of lobeline and its analogs have been reported. However, it is a remarkable fact that no studies have reported the differences in pharmacological activities between the two isomers. In this article, we found that different degrees of isomerization of lobeline injection have significant differences in respiratory excitatory effects in pentobarbital sodium anesthetized rats. Compared with *cis*-lobeline injections, the respiratory excitatory effect was significantly reduced by 50.2% after administration of injections which contained 36.9% *trans*-lobeline. The study on the influencing factors of isomerization between two isomers shown that this isomerization was a one-way isomerism and only converted from *cis* to *trans*, where temperature was the catalytic factor and pH was the key factor. This study reports a new discovery. Despite the widespread use of ventilators, first-aid medicines such as nikethamide and lobeline has retired to second line, but as a nonselective antagonist with high affinity for a4b2 and a3b2 nicotinic acetylcholine receptors (nAChRs). In recent years, lobeline has shown great promise as a therapeutic drug for mental addiction and nervous system disorders, such as depression, Alzheimer disease and Parkinson disease. Therefore, we suggest that the differences between two isomers should be concerned in subsequent research papers and applications.

## 1. Introduction

Isomerism is widespread in organic substances; the discovery of isomerism and its theoretical clarification play an important role in the development of material composition and structural theory [1]. It opens up the study of structural chemistry and promoted the development of organic chemistry. *Cis-trans* isomerism, also known as geometric isomerism, which refers to the diastereoisomerism of the compounds in the molecule due to the restriction of free rotation, is a stereoisomerism in isomerism [2,3]. Generally, *trans* isomers are more stable than *cis* isomers. Isomerization of the *cis*-*trans* isomer [4] is a dynamic equilibrium process which is usually divided into three types: photoisomerization [5], thermoisomerization [6], and catalytic isomerization [7], but is not limited thereto. The differences in physical, chemical, and biological activities due to isomerization are hotspots in organic chemistry research and an important source of new drug discovery.

Lobeline, an alkaloid found in the stems of *Lobelia inflata*, a plant indigenous to North America [8], is white or microstrip yellow crystal or granular powder, odorless, bitter taste and discoloration in case of light and heat. The molecular formula is C_22_H_27_NO_2_, it contains both *cis* and *trans* isomers [9]. The chemical formula and structural formula are shown in Figure 1. The computed properties of *cis*-lobeline, such as XLogP3-AA, hydrogen bond donor count, hydrogen bond accept count, topological polar surface area, were 3.8, 1, 3, and 40.5 Å^2^, respectively. Detailed information could be found in the PubChem database [10].

Evidence suggests that lobeline was a nonselective antagonist with a high affinity for a4b2 and a3b2 nicotinic acetylcholine receptors (nAChRs) [11]. It was used as an emetic and respiratory stimulant in clinical practice since the 19th century and is now prepared by chemical synthesis. As a respiratory stimulant, it is characterized by selective stimulation of the carotid body and aortic chemoreceptors (N-receptors), reflexively exciting the respiratory center, the vagus nerve center, and the vascular movement center, leading to the breath becoming deeper and faster. However, this stimulus lasts only a few minutes. According to the reports in the literature, after intravenous injection, the duration of action usually maintained at 20 min, and the LD_50_ of intravenous injection in mice is 39.9 mg/kg [12]. It was used for neonatal asphyxia and carbon monoxide asphyxiation, anesthetic poisoning such as ether and chloroform, central inhibitors (such as opioids, barbiturates) poisoning, and respiratory failure caused by various diseases such as pneumonia [13] and diphtheria. However, it is not advisable for the respiratory arrest and respiratory weakness caused by progressive respiratory failure.

In recent years, extensive research has shown that lobeline has a multi-directional mechanism of action, including acetylcholine N receptors, opioid receptors, and monoamine transporters. In particular, it has shown great promise as a therapeutic drug for mental addiction and nervous system disorders, such as Alzheimer’s syndrome [14], Parkinson’s syndrome, and depression [15,16]. As a partial nicotine agonist, it has been used in a variety of commercially available preparations to help stop smoking [17].

Despite extensive studies, a comprehensive analysis of the pharmacological activities and the influencing factors of the isomerization of *trans*-lobeline is missing. The presence of *cis* and *trans* isomers in lobeline is well known, and many studies on the relationship between the structure and pharmacological activity of lobeline and its analogs have been reported [18,19,20]. However, the differences in pharmacological activities and the influencing factors of isomerization between the two isomers have not been reported. Due to our limited understanding of *trans*-lobeline, we may underestimate the potential safety risk caused by the *trans* isomer or miss an opportunity to discover a new medicine.

In pharmaceutical research, the potential safety risks and lack of efficacy caused by the simultaneous presence of two or more isomers in a drug have been a challenge for drug quality control. Here, based on the fact that lobeline was used as respiratory stimulants, we investigated the differences in the ability of *cis* and *trans* isomers to excite breathing. The results shown that when the degree of isomerization was 38.7% (the percentage of *trans*-lobeline was 36.9%), the ability to excite breathing decreased by 50.2% compared with *cis*-lobeline. Furthermore, when the degree of isomerization was 67.7% (the percentage of *trans*-lobeline was 22.5%), the ability to excite breathing decreased by 63.0% compared with *cis*-lobeline. It was suggested that the *cis* and *trans* isomers have significant differences in the ability to excite breathing. In addition, considering the necessity for quality control and providing reliable data for pharmacological activity studies, we optimized and established two new HPLC-based methods, which can determine both isomers in a single run for the study of influencing factors of isomerization and the quality evaluation of samples containing lobeline. The results of the isomerization influencing factors experiments showed that pH is the key factor of isomerization, and temperature is the acceleration factor of isomerization. In the solutions containing lobeline, pH should not exceed 2.6 and the temperature in its lifespan should not exceed 40 °C.

## 2. Results

### 2.1. Determination of trans-Lobeline Content

Please see Appendix A. LC-MS/MS for the *trans*-lobeline structure analysis.

#### 2.1.1. Optimized HPLC Conditions Were Feasible

The optimized HPLC conditions enables efficient separation of *cis* and *trans* isomers in processed lobeline hydrochloride solutions (Figure 2). Plate ≥ 8000, both of selectivity, specificity, precision, and accuracy, met the requirements; the extraction and recovery were between 95–105% and the calibrations were linear over a certain range from 30 to 240 ng/mL in all samples with a correlation coefficient (*R*^2^) larger than 0.9999.

#### 2.1.2. Determination of *trans*-Lobeline Percentage Content

It is no possible to quantify the exact content of *trans*-lobeline due to a lack of reference standards. However, based on the area normalization method, it is also acceptable to use the percentage of the peak area to represent the content of each component in the sample. The percentage content of *trans*-lobeline in the three samples described previously are shown in Table 1.

As shown in Table 1, freshly prepared lobeline solution was *trans*-lobeline free; after treated at 105 °C for 15 min, the percentage content of *trans*-lobeline raised to 38.7%, and this value became to 67.7% when it was treated at 121 °C for 120 min.

### 2.2. Effects of Two Isomers on Stimulation of Respiratory Are Significantly Different

The promotion of respiratory rate of anesthetized mice was used to evaluate whether the ability of three samples to excite respiration is different. As shown in Figure 3 and Table 2, the samples processed by 105 °C for 15 min and 121 °C for 120 min have a significantly different ability to stimulation of respiratory compared with the freshly prepared solutions. The freshly prepared solutions increased the respiratory rate by 22.7 ± 4.44, while the samples which processed by 105 °C for 15 min and 121 °C for 120 min only increased by 11.3 ± 2.21 and 8.4 ± 1.07, respectively. The ability of respiratory stimulation decreases as the percentage content of *trans*-lobeline increases.

### 2.3. Method Validation in the Experiment on Influencing Factors of Isomerization

#### 2.3.1. Selectivity and Specificity

As shown in Figure 4, after treatment by all influencing factors involved in this study, there is no interference between the major target matrices (*cis-* and *trans*-lobeline isomers) and newly generated impurities and adjacent peaks reached baseline separation, while the resolution was greater than 1.5. As a part of the method specificity evaluation, peak purity in each sample met the requirements.

#### 2.3.2. Linearity and LLOQ

The standard curve, correlation coefficient of *cis*-lobeline in the selected range and LLOQ were shown in Table 3. The calibrations were linear over a certain range in all samples with a coefficient (*R^2^*) larger than 0.999.

#### 2.3.3. Precision and Accuracy

Obtained accuracy and precision data were summarized in Table 4. The accuracies ranged from 96.25% to 99.67%, along with precision ranging from 0.27% to 0.63%. These data suggested that the method was accurate and reproducible for the quantification of analytes in the samples.

#### 2.3.4. Extraction Recovery

As shown in Table 5, the extraction rate for *cis*-lobeline was ideal, which ranged from 98.38% to 100.51%.

#### 2.3.5. Stability

As mentioned in the previous paragraph, lobeline was unstable to heat, light, acid, and base, and its stability was conditional. Among the influencing factors of the isomerization experiments, low temperature (2–8 °C) and pH (do not exceed 3.0) were proven to have no effect on the stability of *cis*-lobeline in 60 days.

### 2.4. Influencing Factors of Isomerization

#### 2.4.1. Temperature Is the Catalytic Factor of Isomerization

As shown in Figure 5 and Table 6, *cis*-lobeline solution, which placed at 2–8 °C for 60 days, did not generated any impurities and the content was stable. Placed at 40 °C for 60 days, impurities were significantly increased due to the conversion of *cis*-lobeline to *trans*-lobeline. However, when *cis*-lobeline solution was treated at 121 °C for 120 min, most of the *cis*-lobeline was destroyed, part of it was converted to *trans*-lobeline, and the other part became unknown impurities.

#### 2.4.2. pH Is the Key Factor of Isomerization

As shown in Figure 6 and Figure 7 and Table 7, the stability of *cis*-lobeline decreased with increased pH. The higher the pH, the easier it was converted to *trans*-lobeline. When the pH was 2.6, it was an inflection point of isomerization.

According to the test of influencing factors of isomerization, we can conclude that pH is the key factor of isomerization, as it determines the final content of *trans*-lobeline. The *cis*-lobeline solution formulated with low pH solvent can withstand high temperature-induced catalysis of isomerization. Temperature is the catalytic factor of isomerization; high temperature can either catalyze the isomerization process or it can destroy *cis*-lobeline directly.

## 3. Materials and Methods

### 3.1. Drugs and Reagents

*Cis*-lobeline chemical reference substance (CRS): Sigma-Aldrich (LOT: 14109JBV). Pentobarbital was purchased from Sigma-Aldrich (CAS: 57-33-0). Water was prepared by double distillation. HPLC-grade acetonitrile was supplied by Fisher Scientific Company Inc. (Boston, MA, USA). All other reagents were analytical grade and purchased from Sinopharm Chemical Reagent Co., Ltd. (Beijing, China).

### 3.2. Animals, Animal Management, and Experimental Design

Pharmacodynamic study of *trans*-lobeline isomers was conducted in the apparently healthy adult Wistar rats (180–200 g, male). Animals were purchased from the Beijing Keyu Animal Breeding Center (Production license: SCXK (Jing)-2010-0010). All animals were housed in an environmentally controlled breeding room (temperature: 22 ± 2 °C, humidity: 50 ± 5%, dark/light cycle: 12/12 h). The animals were provided standard laboratory food and water. The experimental protocols were performed in accordance with the guidelines of the National Institutes of Health for the Care and Use of Laboratory Animals. Prior to each experiment, all animals were kept under laboratory conditions for a period of 3 days or more for acclimatization.

### 3.3. Determination of cis-Lobeline and trans-Lobeline Content

Due to the lack of lobeline (both *cis* and *trans* isomer) reference standards, we cannot accurately determine the absolute content of these two isomers. However, based on the area normalization method, it is also an acceptable way to use the percentage of the peak area to represent the content of each component in the samples, and *cis*-lobeline CRS allows us to measure the content of *cis*-lobeline in the sample relatively accurately. Therefore, a HPLC method was optimized in order to relative quantify the content of *cis*-lobeline and *trans*-lobeline in the samples to be tested in the pharmacodynamic study of *trans*-lobeline isomers.

Chromatographic analysis was performed on an Agilent 1100 Series (Agilent Technologies, St. Clara, CA, USA) LC system containing a quaternary pump, an online degasser, an autosampler, and a thermostatic column compartment set at 30 °C. Chromatographic separation was conducted on a Waters SymmetryShield RP18 column (5 μm, 250 × 4.6 mm) (Milford, MA, USA). Elution was carried out by isocratic elution. The mobile phase consisted of a mixture of 50 mM phosphate buffer (pH 3.0) (A) and acetonitrile (B), where the ratio of A was 80%. The flow rate was 1.0 mL/min. UV detection wavelength was 246 nm.

Our method was different from the isocratic elution method used in the pharmacodynamic study. For more sensitive detection of the impurities in the samples to be tested in the isomerization influencing factors experiment, a new gradient elution method was established. This new method has higher detection rate and lower limits of quantification (LLOQ). Chromatographic analysis was performed on the same Agilent 1100 Series LC system. The mobile phase consisted of a mixture of 50 mM phosphate buffer (pH 3.0) (A) and acetonitrile (B). Gradient elution was performed as follows: (1) mobile phase A was at 85% at 0 min, (2) an isocratic elution was maintained at 85% A from 0 min to 10 min, (3) a linear gradient was decreased to 82% A from 10 min to 30 min, and (4) a linear gradient was decreased to 72% A from 30 min to 70 min. The flow rate was 1.0 mL/min. UV detection wavelength was 210 nm.

### 3.4. Method Validation

The method used to determine content in pharmacodynamic studies was adopted from reference [21]. In the article, the mobile phase consisted of a mixture of 50 mM phosphate buffer (pH 3.0) (A) and methanol (B), where the ratio of A was 55%. UV detection wavelength was 210 nm. However, in this manuscript, since methanol has terminal absorption at 210 nm, which will affect the accuracy of content determination, acetonitrile was used to replace methanol. Moreover, the proportion of acetonitrile was adjusted to 20% in order to make *cis*-lobeline and *trans*-lobeline better separated. The adjustment of the method was slight and there was no method inapplicability in the subsequent content determination. We did not perform method validation as we just optimized the pH and ratio of mobile phase. However, we performed method validation in the experiment on influencing factors of isomerization because of a big change in approach. These parameters include selectivity and specificity, linearity and LLOQ, precision and accuracy, extraction recovery and stability.

### 3.5. Pharmacodynamic Study of trans-Lobeline Isomer

The experiments were performed on Wistar rats, anesthetized with sodium pentobarbital solution (60 mg/kg) via the tail vein. The anesthetized rats were fixed on multi-channel electrophysiological apparatus (RM6240E, Chengdu Instrument Factory, Chengdu, China), and their respiratory rate and heart rate were observed and recorded. According to the changes in the respiration rate, pentobarbital sodium (0.1 mL) was injected every 1–2 min until the respiration rate was reduced to 50 bpm and maintained for 3–5 min. Respiration rate and heart rate were collected every 5 min. In 45 min, all rats were injected with injectable freshly prepared lobeline hydrochloride solution, lobeline hydrochloride solution sterilized for 15 min at 105 °C, and lobeline hydrochloride solution sterilized for 120 min at 121 °C at a dosage of 2 mg/kg. Heart rate and respiratory rate were recorded at an interval of 5 min. Mean respiratory rate five minutes before and after administration and elevated respiratory rate after administration were calculated for each rat.

### 3.6. Study on the Influencing Factors of Isomerization

As lobeline is unstable to heat, light, acid, and base, studies on the isomerization influencing factors were focused on temperature and Pondus Hydrogenii (pH).

#### 3.6.1. Investigation of the Relationship between Temperature and Isomerization

We mainly investigated sterilization temperature and storage temperature. *cis*-Lobeline CRS (*trans*-lobeline free) was dissolved in water and pH was adjusted to 3.0 by using 0.1 mol/L hydrochloric acid. Then, we treated these samples and collected data according to the study overview, which is shown in Figure 8. The prepared samples were diluted with mobile phase into a solution containing 120 μg/mL of *cis*-lobeline for HPLC analysis.

#### 3.6.2. Investigation of the Relationship between pH and Isomerization

Hydrochloric acid solutions of different pH were prepared, and the pH was determined to be 1.97, 2.04, 2.18, 2.32, 2.64, 2.87, 2.97, 3.10, 3.24, 3.43, 3.63, 3.90, and 4.27. A total of 3 mg/mL *cis*-lobeline hydrochloride solution was prepared by using the different pH solutions as solvent, and sealed by melting the glass ampoules. To improve experiment efficiency, 105 °C and 15 min autoclaving was used to accelerate the isomerization process. The prepared samples were diluted with mobile phase into a solution containing 120 μg/mL of lobeline for HPLC analysis.

### 3.7. Statistical Analysis

SAS version 9.2 (SAS Inc., Cary, NC, USA) was used to perform the statistical analyses. All data were expressed as the mean ± SD. For comparisons between two groups, Student’s t-test was performed. For comparisons among three or more groups, the data were analyzed using a one-way analysis of variance (ANOVA). For all analyses, a *p*-value of < 0.05 was considered to indicate statistical significance, and a *p*-value of < 0.01 was considered to indicate extreme statistical significance. GraphPad prism (version 8.0, GraphPad Software, San Diego, CA, USA) was used to draw the statistical figures.

## 4. Discussion

As described in the results, the isomerization of lobeline is affected by temperature and pH. The higher the temperature, the higher the degree of isomerization, and the lower the pH, the lower degree of isomerization. Among the existing literatures that can be searched, there is no similar article describing the factors of lobeline isomerization. However, as described in the introduction section, thermoisomerization is one of the types of transformation of *cis-trans* isomers. Isomerization due to temperature changes is a very common phenomenon [22,23]. As shown in Figure 3, after being treated by different temperatures, there are significant differences in the excitatory effects of lobeline solutions with different degrees of isomerization on respiration. The data show that when the degree of isomerization is 38.7%, the respiratory stimulation effect is only half of that of fresh solution, and when the degree of isomerization is 67.7%, the respiratory stimulation effect is only 37% of that of fresh solution.

However, 105 °C and 121 °C are the sterilization processes that were often used in lobeline hydrochloride injection production. We further detected the content of *cis*- and *trans*-isomers in lobeline hydrochloride injections which are currently commercially available; the results also confirmed that both isomers were detected simultaneously. However, the samples from different companies have slightly different degrees of isomerization. The above results suggest that lobeline hydrochloride injection has defects in the efficacy of excitatory breathing. We also hope to provide a warning for the use of lobeline through this manuscript, and hope that users know and will manage this risk.

Another type of transformation of *cis-trans* isomers is catalytic isomerization. Noble metal catalysts supported by a solid acid, such as platinum-alumina, platinum-molecular sieve, palladium-alumina, etc., are common catalysts for catalytic isomerization [24]. Such catalysts consist of bifunctional catalysts in which the metal component acts as a hydrogenation and dehydrogenation and the solid acid acts as an isomerization. When such catalysts were used, the reaction needs to be carried out in the presence of hydrogen, so it is also called a hydroisomerization catalyst. However, these catalysts were often used for gas phase isomerization.

There are also differences in the pharmacological activities of the two isomers for respiratory excitatory effects. However, unfortunately, there is currently no pure *trans*-lobeline, so it is impossible to directly compare the differences in the pharmacological activities of the two isomers. An indirect comparison can only be made through the *trans*-lobeline produced by isomerization. For lobeline hydrochloride injection, the difference in isomer activity is a side effect. However, for those two isomers alone, this is an opportunity. Analogs [11] such as *meso*-Transdiene (MTD), lobelane, *nor*-lobelane, carboxylic acid, and sulfonic acid ester are being investigated currently as a clinical candidate for the treatment of psychostimulant abuse [25]. These analogs might lack nicotinic receptor affinity [24], retain affinity for vesicular monoamine transporter 2 (VMAT2) [20], or enhance affinity for dopamine (DA) and serotonin transporters. In summary, for chemical receptors that interact with lobeline, the specificity of activity between different analogs were inconsistent, which provides a reference for the new clinical application study of *trans*-lobeline.

How to carry out future research is something we have been considering. Here, we believe that the following issues should be focused on. First, the proper mechanism by which pH induced isomerization should be revealed. Secondly, the study of isomer activity should not be limited to nAChRs receptors, nor should it be restricted to the excitatory respiration effect [26]. Last but not the least, the study of the structure–activity relationship of *trans*-lobeline should be considered in subsequent studies.

## Figures and Tables

**Figure 1 molecules-27-06253-f001:**
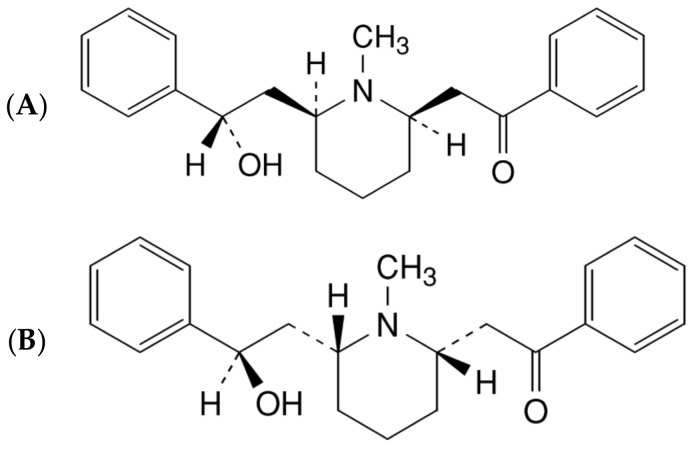
The chemical structure of *cis*-lobeline ((−)-lobeline) and *trans*-lobeline ((+)-lobeline). (**A**): 2-[(2*R*,6*S*)-6-[(2*S*)-2-Hydroxy-2-phenylethyl]-1-methylpiperidin-2-yl]-1-phenylethanone ((−)-lobeline); (**B**): 2-[(2*S*,6*R*)-6-[(2*R*)-2-hydroxy-2-phenylethyl]-1-methylpiperidin-2-yl]-1-phenylethanone ((+)-lobeline); Molecular formula: C_22_H_27_NO_2_, MW:337.

**Figure 2 molecules-27-06253-f002:**
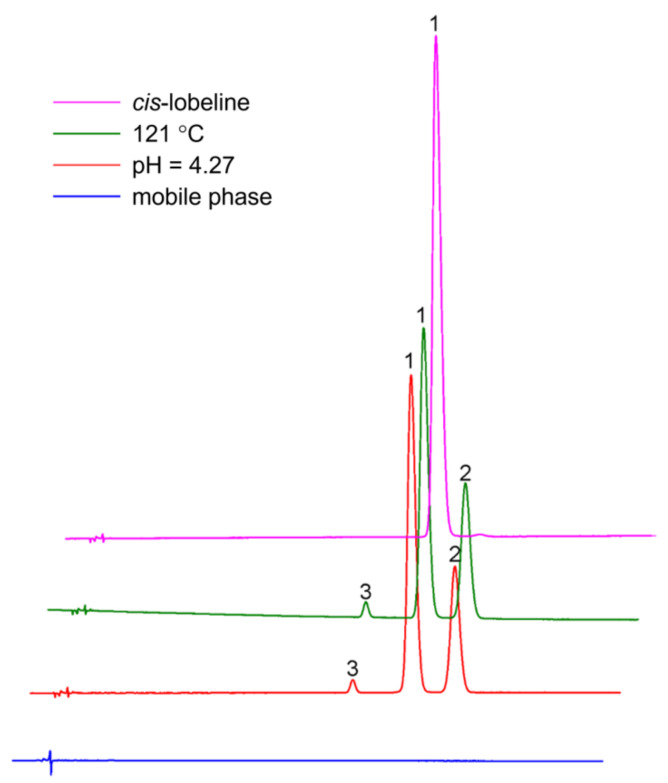
Representative chromatograms to evaluate the separation effect of isocratic elution methods. 1 = *cis*-lobeline, 2 = *trans*-lobeline, 3 = Intermediate.

**Figure 3 molecules-27-06253-f003:**
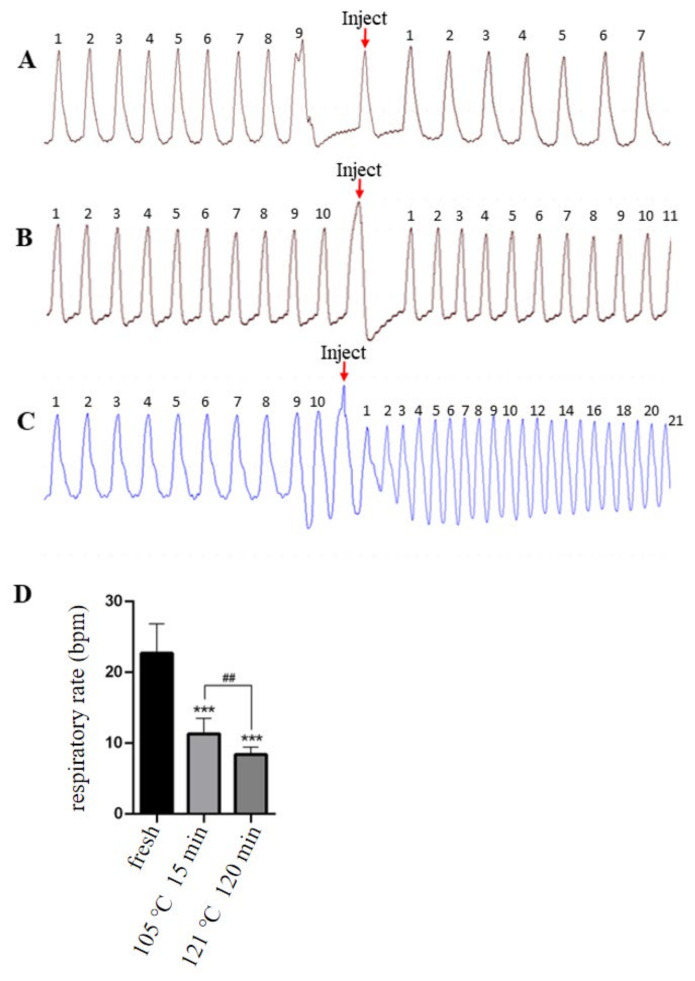
Effects of lobeline injection with different isomerization levels on excited respiration in rats with respiratory depression. (**A**): treated at 121 °C for 120 min, (**B**): treated at 105 °C for 15 min, (**C**): fresh, (**D**): respiration rate. “***” represents compared with the Fresh group, *p* ≤ 0.001, “^##^” represents compared with the 105 °C 15 min group, *p* ≤ 0.01.

**Figure 4 molecules-27-06253-f004:**
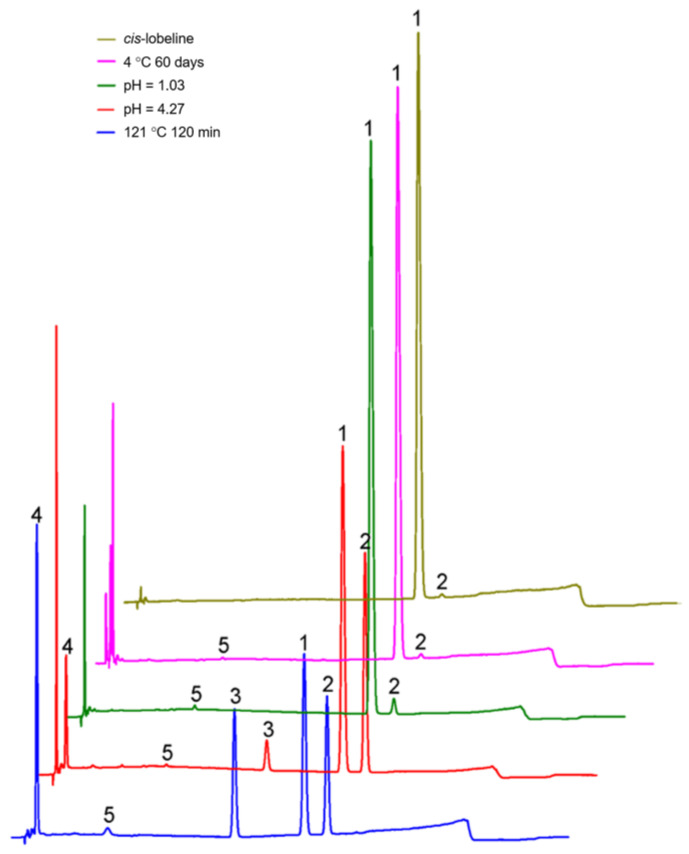
Representative chromatograms to evaluate the separation effect of gradient elution methods. 1 = *cis*-lobeline, 2 = *trans*-lobeline, 3 = Intermediate, 4 and 5 = unknown impurities.

**Figure 5 molecules-27-06253-f005:**
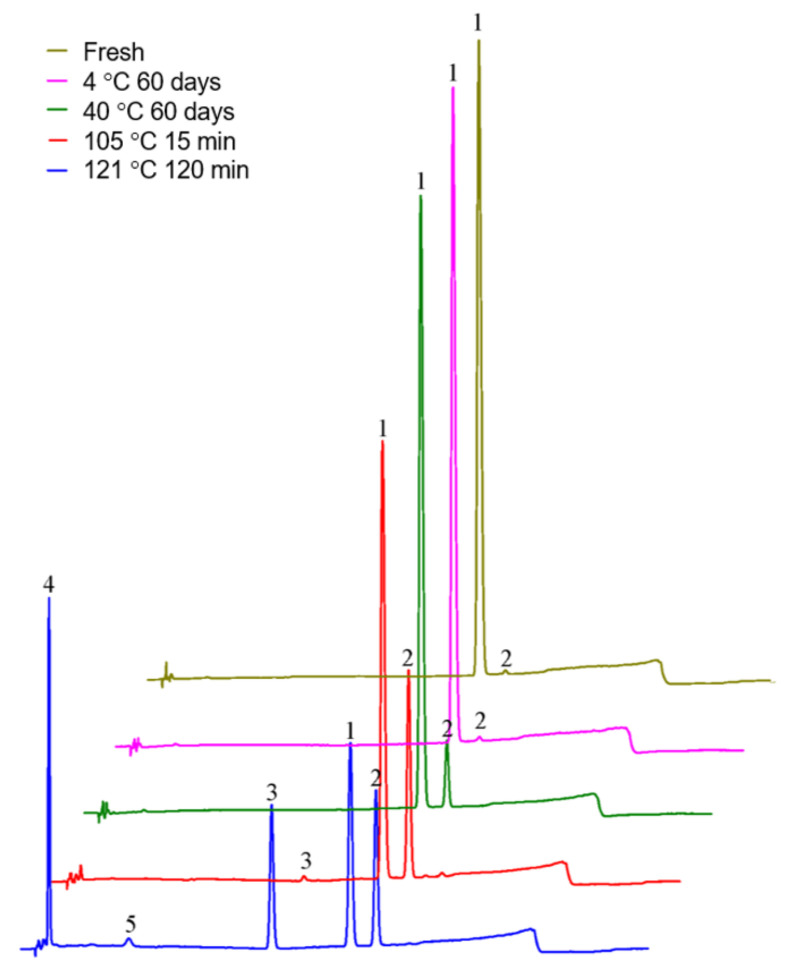
Formation of impurities in the *cis*-lobeline reference solution after it was treated by different temperature. 1 = *cis*-lobeline, 2 = *trans*-lobeline, 3 = Intermediate, 4 and 5 = unknown impurities.

**Figure 6 molecules-27-06253-f006:**
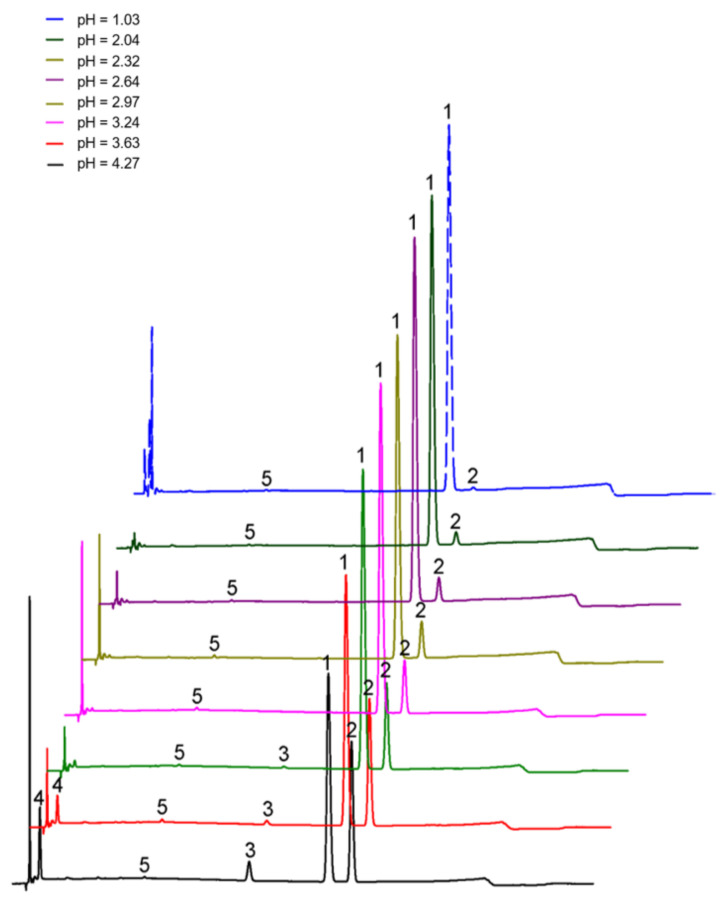
Formation of impurities in the *cis*-lobeline reference after it was processed by different pH solutions and autoclaving at 105 °C for 15 min. 1 = *cis*-lobeline, 2 = *trans*-lobeline, 3 = Intermediate, 4 and 5 = unknown impurities.

**Figure 7 molecules-27-06253-f007:**
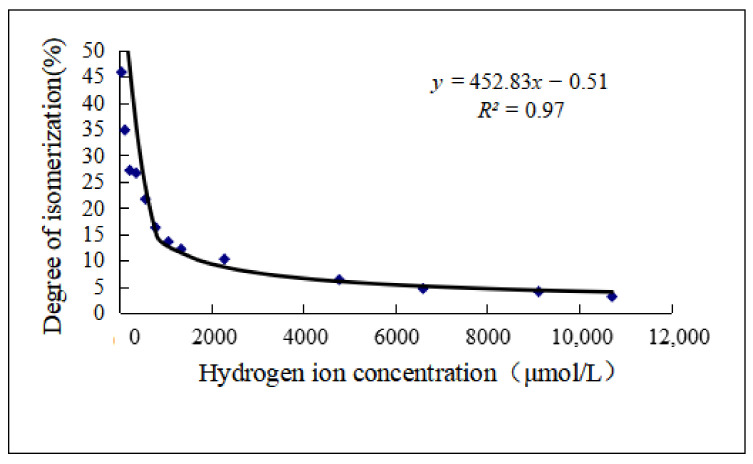
Relationship between pH and degree of isomerization (*n* = 3).

**Figure 8 molecules-27-06253-f008:**
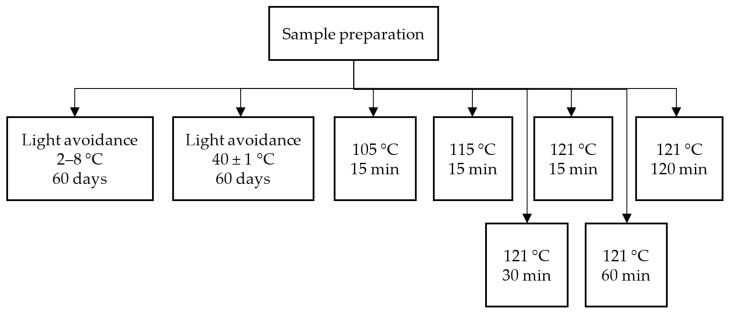
The experiment program of sample preparation in the temperature and isomerization test.

**Table 1 molecules-27-06253-t001:** The percentage content of *trans*-lobeline in the samples used to pharmacodynamic study (*n* = 3).

Sample Name	*cis*-Lobeline (%)	*trans*-Lobeline (%)	Degree of Isomerization (%)
Fresh	100%	0	0
105 °C 15 min	61.3 ± 0.45%	36.9 ± 0.68%	38.7%
121 °C 120 min	32.3 ± 0.47%	22.5 ± 1.05%	67.7%

Note: Degree of isomerization = 1 − Average of *cis*-lobeline (%).

**Table 2 molecules-27-06253-t002:** Relationship between increased respiratory rate and degree of isomerization (*n* = 10).

Sample Name	Degree of Isomerization (%)	Increased Respiratory Rate (bpm)
Fresh	0	22.7 ± 4.14
105 °C 15 min	38.7%	11.3 ± 2.21 ***
121 °C 120 min	67.7%	8.4 ± 1.07 ***^, ##^

Note: “***” means compared with the Fresh group, *p* ≤ 0.001, “^##^” means compared with the 105 °C 15 min group, *p* ≤ 0.01.

**Table 3 molecules-27-06253-t003:** Standard curves, correlation coefficients, linear range, and LLOQ of *cis*-lobeline.

Name	Standard Curves	*R* ^2^	Linear Range (μg/mL)	LLOQ (μg/mL)
*cis*-lobeline	*y* = 21.426*x* − 8.0696	0.9998	30–240	0.36

**Table 4 molecules-27-06253-t004:** Intra-/interday accuracy and precision of the developed assay (*n*= 6).

Name	QC Conc. (μg/mL)	Intraday			Interday		
		Calc Conc. (μg/mL)	RSD (%)	Accuracy (%)	Calc Conc. (μg/mL)	RSD (%)	Accuracy (%)
*cis*-lobeline	60.52	60.08 ± 2.13	0.27	99.67 ± 0.28	61.03 ± 1.96	0.38	99.15 ± 1.06
	121.48	120.61 ± 3.27	0.50	98.79 ± 1.02	120.57 ± 2.69	0.50	98.76 ± 0.94
	242.55	245.37 ± 4.95	0.69	97.34 ± 2.06	240 ± 3.54	0.63	96.25 ± 3.21

**Table 5 molecules-27-06253-t005:** Recovery of the developed assay (*n* = 9).

Group	Sample Intake (mg)	Reference Material Added (mg)	Content Determination (mg)	Recovery (%)	Average (%)	RSD (%)
80%	2.122	1.960	4.016	98.37	98.38	0.51
	2.122	1.960	3.997	97.89		
	2.122	1.960	4.037	98.89		
100%	2.653	2.450	5.119	100.31	100.51	0.61
	2.653	2.450	5.165	101.20		
	2.653	2.450	5.105	100.03		
120%	3.184	2.941	6.138	100.23	100.27	0.42
	3.184	2.941	6.117	99.88		
	3.184	2.941	6.168	100.71		

**Table 6 molecules-27-06253-t006:** Relationship between temperature and degree of isomerization (*n* = 3).

Sample Name	*cis*-Lobeline (%)	*trans*-Lobeline (%)	Degree of Isomerization (%)
Fresh	100	0	0
4 °C 60 days	99.2 ± 0.39	0.6 ± 0.08	0.8
40 °C 60 days	93.2 ± 0.63	6.7 ± 0.49	6.8
105 °C 15 min	61.3 ± 0.45	36.9 ± 0.68	38.7
121 °C 120 min	32.3 ± 0.47	22.5 ± 1.05	67.7

Note: Degree of isomerization = 1 − Average of *cis*-lobeline (%).

**Table 7 molecules-27-06253-t007:** Relationship between pH and degree of isomerization (*n* = 3).

pH	Hydrogen Ion Concentration (μmol/L)	Degree of Isomerization (%)
1.97	10,715.2	3.084
2.04	9120.1	4.04
2.18	6606.9	4.629
2.32	4786.3	6.316
2.64	2290.9	10.237
2.87	1349.0	12.132
2.97	1071.5	13.537
3.1	794.3	16.236
3.24	575.4	21.66
3.43	371.5	26.61
3.63	234.4	27.134
3.9	125.9	34.796
4.27	53.7	45.824

Note: Degree of isomerization = 1 − Average of *cis*-lobeline (%).

## Data Availability

The datasets generated and analyzed in the current study are available from the corresponding author upon reasonable request.

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
