# Peer review of "From cis-Lobeline to trans-Lobeline: Study on the Pharmacodynamics and Isomerization Factors"

_molecules, 2022, doi:10.3390/molecules27196253_

Round 1

Reviewer 1 Report

The work is about the isomerization of cis-Lobeline to trans-Lobeline: Study on the Pharmacodynamics and Isomerization Factors. While the work is interesting, there are several issues, including writing, experimentation, rationalization, and discussion. For instance, While the authors found that pH has an impact on isomerization, it would be better if you came up with a proper mechanism. In terms of the analytical method, I am wondering how the authors confirmed the trans isomer without any ambiguity. I am wondering if you use the mass spectrometry approach to confirm the isomer accurately with not only m/z but also fragmentation patterns. Moreover, the authors should also provide more details on the analytical method, validation, and sample analysis.

Reviewer 2 Report

The manuscript describes the study on lobeline cis-trans isomerism. Isomerism finds its importance in the field of clinical pharmacology and pharmacotherapeutics. It's worth to notice that isomers differ in their pharmacokinetic and pharmacodynamic properties. Drug isomerism has opened a new era of drug development. Currently, knowledge of isomerism has helped us in introducing safer and more effective drug alternatives of the newer as well as existing drugs. It is a remarkable fact that no studies have reported the differences in pharmacological activities between the two isomers of lobeline.

After reading the manuscript in detail, I suppose that the all paragraphs together describe the presented study in broad and appropriate way. The conclusions presented by the authors are consistent with the evidence and relate to the main research issue. However, a major drawback of the work is lack of the study limitations. In addition, I recommend to extend the chapter "2.4 Method Validation". Reference [19] is not generally available in global databases. In my opinion, in discussion the potential effects of lobeline thermal decomposition products on rats should be carried out. I missed more references in discussion and comparing the obtained results with literature data. Additionally, the small number of citations of works from the last 5 years is rather surprising (out of 25 bibliographic items, only 8 works have been published since 2017). Authors need to conduct detailed survey of the literature and supplement the citation with references to recent studies in the field.

To enrich the manuscript I also recommend as follows:
- Molecules journal is dedicated to molecules. I suggest that introduction section should be improved by including the chemical background of lobeline (e.g. chemical properties such as: logP, hydrogen bonds donor and acceptors or polar surface area). It can be  presented in the form of a short paragraph or table within the introduction section
- Figure 4 needs to improve (graph D has unnatural proportions and is uneasy to read)

Round 2

Reviewer 2 Report

The authors properly revised the manuscript according to my comments and I have no further questions.